# CorBenchX: Large-Scale Chest X-Ray Error Dataset and Vision–Language Model Benchmark for Report Error Correction

## Abstract

AI-driven models have shown great promise in detecting errors in radiology reports, yet the field lacks a unified benchmark for rigorous evaluation of error detection and further correction. To address this gap, we introduce **CorBenchX**, a comprehensive suite for automated error detection and correction in chest X-ray reports, designed to advance AI-assisted quality control in clinical practice. We first synthesize a large-scale dataset of 26,326 chest X-ray error reports by injecting clinically common errors via prompting DeepSeek-R1, with each corrupted report paired with its original text, error type, and human-readable description. Leveraging this dataset, we benchmark both open- and closed-source vision–language models (*e.g.*, InternVL, Qwen-VL, GPT-4o, o4-mini, and Claude-3.7) for error detection and correction under zero-shot prompting. Among these models, o4-mini achieves the best performance, with 50.6 % detection accuracy and correction scores of BLEU 0.853, ROUGE 0.924, BERTScore 0.981, SembScore 0.865, and CheXbertF1 0.954, remaining below clinical-level accuracy, highlighting the challenge of precise report correction. To advance the state of the art, we propose a multi-step reinforcement learning (MSRL) framework that optimizes a multi-objective reward combining format compliance, error-type accuracy, and BLEU similarity. We apply MSRL to QwenVL2.5-7B, the top open-source model in our benchmark, achieving an improvement of 38.3% in single-error detection precision and 5.2% in single-error correction over the zero-shot baseline.

## 1 Introduction

In modern clinical practice, radiology examination is indispensable, and the demands are increasing due to aging populations, broader imaging recommendations in updated clinical guidelines, and the increasing availability of equipment Afshari Mirak et al. (2025). As demand surges, radiologists face escalating workloads, which in turn heightens the risk of diagnostic errors in radiology reports Kim et al. (2025). To alleviate diagnostic errors in radiology reports, general healthcare systems employ a two-tiered reporting workflow: resident physicians draft preliminary reports that are reviewed, corrected, and finalized by board-certified radiologists Gertz et al. (2024). While this hierarchical process improves accuracy, it demands extensive human resources and is time-consuming. Despite such efforts, diagnostic errors, including misdiagnoses, missed diagnoses, and delayed diagnoses, occur at rates as high as 10–26% Zhang et al. (2023); Pesapane et al. (2024). These errors not only pose serious threats to patient safety and impose substantial economic burden but also increase the likelihood of malpractice suits against radiologists Kasalak et al. (2023).

Given these persistent challenges, there is increasing interest in leveraging Large language Models (LLMs) to streamline radiology reporting and reduce human burden—yet current approaches face critical limitations. Recent advances in LLMs have catalyzed interest in automated radiology report generation Chen et al. (2024; 2023); Tanno et al. (2025); Lang et al. (2025). LLM-driven systems can draft impressions and suggest follow-up recommendations, promising to alleviate radiologists' workload. However, despite their fluency, these generative approaches often fall short of clinical-grade reliability. Common issues such as hallucinated findings, formatting inconsistencies, and domain-mistranslations remain prevalent Zeng et al. (2024), necessitating extensive human oversight and limiting their integration into real-world clinical workflows. *In contrast to the majority of prior*

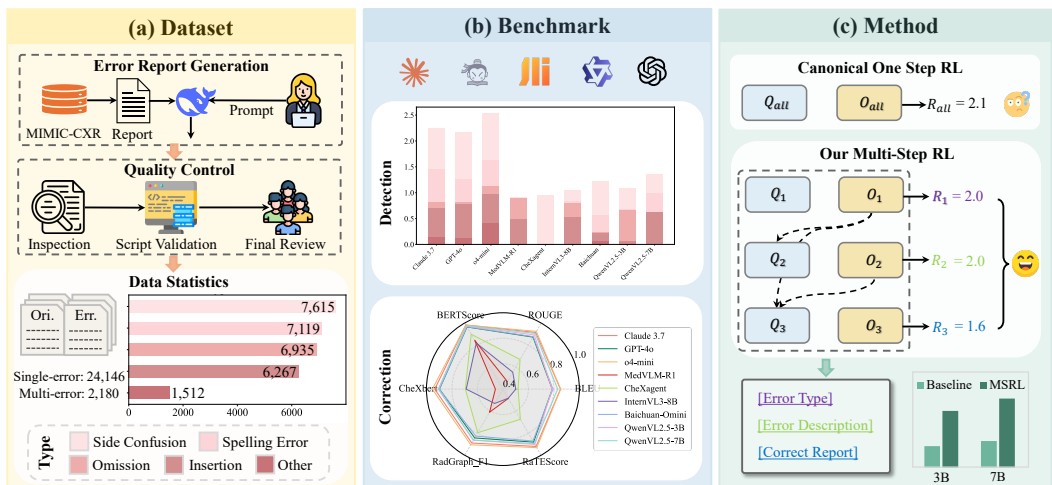

Figure 1: **Overview of CorBenchX.** (a): Error report dataset construction pipeline and dataset statistics. (b): Benchmark results across nine vision–language models for error detection and correction. (c): Illustration of our proposed multi-step reinforcement learning (MSRL) method and its performance improvements over the baseline.

*work on generating reports, we shift the emphasis toward automated error detection and correction in radiology reports, which is a critical yet underexplored task in radiology AI.*

Recent studies have demonstrated the potential of LLMs for automated error detection in radiology reports Gertz et al. (2024); Kim et al. (2025); Salam et al. (2025); Yan et al. (2025), and several specialized error datasets have been introduced, such as ReXVal Yu et al. (2023a) and RadEvalX Calamida et al., RRED Min et al. (2022), and ReXErr Rao et al. (2024). However, these efforts exhibit critical limitations: 1) most evaluations rely on small, manually curated corpora that fail to represent the full diversity of clinical reporting mistakes; 2) they focus exclusively on error detection, offering no end-to-end correction; and 3) many datasets are either not publicly accessible or omit clinically common error types such as laterality confusion. Moreover, there is currently no unified benchmark that evaluates both detection and correction across a large-scale, systematically constructed dataset.

To address these gaps, we introduce CorBenchX, a comprehensive benchmark for error detection and correction in chest X-ray reports. As illustrated in Figure 1, we first construct a novel and large-scale chest X-ray error dataset derived from the MIMIC-CXR dataset Johnson et al. (2019) by injecting clinically common mistakes via DeepSeek-R1 prompting. Then we rigorously benchmark nine open- and closed-source VLMs for error detection and correction under zero-shot prompting. Finally, we propose a multi-step reinforcement learning method that optimizes for format compliance, error-type accuracy, and textual fidelity, yielding substantial improvements (38.3% in detection and 5.2% in error correction) over the baseline model. To sum up, our contributions are threefold:

- We present CorBenchX, a large-scale dataset comprising 26,326 chest X-ray error reports, including 24,146 single-error and 2,180 multi-error cases, each annotated with error spans, error type, and concise descriptions.

- We conduct extensive evaluations on the error dataset with various open and closed-source VLMs for both single-error and multi-error detection and correction. The results reveal that current VLMs, while powerful, fall short of meeting the clinical precision required for reliable error detection and correction in radiology reports.

- We propose a novel multi-step reinforcement learning framework to enhance the VLMs via sequential reasoning for error detection, description, and correction.

## 2 RELATED WORKS

### 2.1 RADIOLOGY REPORT GENERATION AND EVALUATION

Automated radiology report generation has rapidly evolved. Early encoder–decoder frameworks combined convolutional or transformer-based image encoders with BERT-style decoders to directly translate image features into narrative reports Syeda-Mahmood et al. (2020); Wang et al. (2022). Retrieval-based approaches, such as MedWriter Yang et al. (2021), which incorporated a hierarchical retrieval mechanism and a hierarchical-LSTM decoder to generate the report by fusing the features from the previous modules. CXR-RePaiR Endo et al. (2021), leverage pre-trained contrastive image–text embeddings to retrieve the most similar reports from a large corpus and adapt them to new cases. More recently, large-scale vision–language pretraining has enabled great progress in the automatic report generation, such as CheXagent Chen et al. (2024), LLM-CXR Lee et al. (2023), and VLCI_MIMIC Chen et al. (2023). Furthermore, ReXrank Zhang et al. (2024) provides a public leaderboard for report generation evaluation, where 8 metrics are adopted as evaluation metrics.

Previous evaluation of generated reports has largely depended on lexical similarity (*e.g.*, ROUGE-L Lin (2004), BLEU Papineni et al. (2002)), which often fail to capture subtle but clinically meaningful edits. To address this, entity–centric measures have emerged: CheXbert F1 Smit et al. (2020) assesses agreement in disease labels inferred from text, while RadGraph-F1 Yu et al. (2023b) evaluates the accuracy of extracted entity–relation graphs that encode findings and anatomical locations. Recently, LLM-related metrics like GREEN Ostmeier et al. (2024) use LLM for error annotation, yielding both quantitative scores and qualitative explanations of clinically significant mistakes.

### 2.2 REPORT ERROR DETECTION

LLMs have recently been applied to detecting errors in radiology reports. Gertz *et al.* Gertz et al. (2024) evaluated GPT-4 on 100 chest X-ray reports with synthetically introduced errors, reporting an average detection accuracy of 82.7%, which surpassed radiology residents (80.0%) but remained below senior radiologists. Similarly, Kim *et al.* Kim et al. (2025) injected interpretive and factual errors into 300 reports, finding that GPT-4 achieved 84% accuracy on interpretive errors and 89% on factual errors. Salam *et al.* Salam et al. (2025) evaluated open-source (Llama 3-70B, Mixtral 8x22B) and closed-source (GPT-4o) models, with GPT-4o significantly outperforming others. Yan *et al.* Yan et al. (2025) extended error detection to Chinese ultrasound reports, evaluating 400 reports with 243 annotated errors; Claude 3.5 Sonnet achieved the highest detection rate. Although these studies underscore the potential of LLMs for automated report review, they exhibit key limitations: existing works rely on small, manually curated error sets that may not capture the full spectrum of clinically errors; most works focus solely on error detection without correction, limiting their practical utility; and many approaches depend on human-in-the-loop validation, which restricts scalability in high-throughput clinical environments.

### 2.3 ERROR DETECTION DATASET

Several datasets have been introduced for radiology report error detection. Yu *et al.* Yu et al. (2023a) proposed the ReXVal dataset, which includes 200 AI-generated/ground-truth report pairs that six radiologists evaluated for clinically significant versus insignificant errors. RadEvalX Calamida et al. comprising 74 chest X-ray reports generated by an M2Tr model on IU-Xray cases, each meticulously annotated by expert radiologists for the presence and clinical severity of reporting. RRED Min et al. (2022) utilized a "generator" to generate findings-impression inconsistent errors in MIMIC-CXR reports and supplemented this with manual error annotations by two radiologists on 111 cases. Sun *et al.* Sun et al. (2025) generated 1,656 chest X-ray reports using GPT-4. Half were error-free; the other half contained errors introduced via prompts. Meanwhile, an additional set of 307 real MIMIC-CXR reports was paired with 307 GPT-4 versions containing errors. While these datasets offer valuable insights into error analysis and detection, their small scale (no more than 200 cases in ReXVal and RadEvalX) and limited public accessibility (RRED and Sun's dataset) hinder their suitability for large-scale evaluation. ReXErr Rao et al. (2024) delivers a public large-scale dataset for chest X-Ray error detection. However, its uniform injection of exactly three errors per report fails to mirror real-world error distributions, omits critical categories such as laterality confusion, and risks introducing internally contradictory mistakes. Moreover, ReXErr does not include standardized error detection and correction benchmarks.

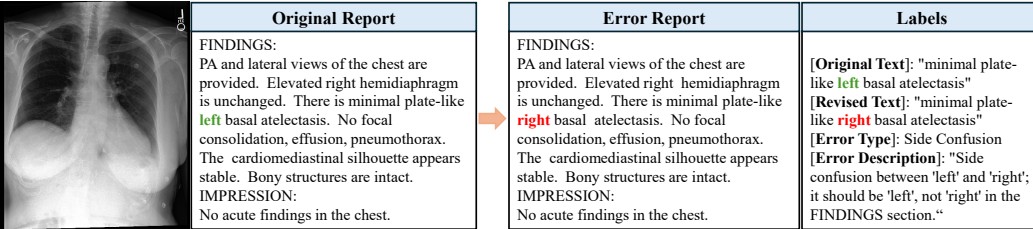

Figure 2: Example of a chest X-ray, paired original radiology report, and the corresponding error-injected report with labels. Text spans highlighted in red denote the injected errors, while the corrected spans are shown in green.

Table 1: Error type explanation and data statistics

| Error type | Description | Number of error instances |
|---|---|---|
| Omission | Missing relevant clinical findings or words | 6,267 |
| Insertion | The unintentional insertion of incorrect words or expressions | 6,935 |
| Spelling Error | Spelling mistakes or typos | 7,119 |
| Side Confusion | Errors involving side or orientation | 7,615 |
| Other | Mistakes in units of measurement, punctuation mistakes, etc. | 1,512 |
| Total | - | 29,448 |

## 3 ERROR REPORT DATASET CONSTRUCTION

We introduce **CorBenchX**, a high-quality and systematically constructed dataset for chest X-ray report error detection and correction. The dataset simulates realistic reporting errors across a range of clinically motivated categories, providing a reliable foundation for training and evaluation.

**Dataset Source and Sampling.** CorBenchX is built on the publicly available MIMIC-CXR dataset Johnson et al. (2019), which contains de-identified chest X-ray reports collected from Beth Israel Deaconess Medical Center. We extract the "Findings" and "Impression" sections from each report and remove records where both sections are empty. From the resulting pool, we randomly sample 26,326 clean reports as the basis for synthetic error injection.

**Error Injection Procedure.** To create realistic errorful variants, we use the DeepSeek-R1 API with a carefully designed prompt (see Appendix B.2 for details). Each API call outputs an error-injected report along with: (i) the error type label, (ii) the paired original and altered text spans, and (iii) a concise natural language error description (see Figure 2). Analysis of 300 real-world radiology reports corrected by senior radiologists at our collaborative institution revealed that single-error cases outnumber multi-error cases by approximately 10:1. To simulate this ratio, we introduce structured perturbations into clean reports in two types of samples:

- **Single-error reports:** Each report contains exactly one error from one of five categories—*omission*, *insertion*, *spelling error*, *side confusion*, or *other*—resulting in 24,146 single corrupted samples.

- **Multi-error reports:** To better reflect real-world reporting complexity, we additionally generate 2,180 reports containing two to three independent errors.

**Quality Control Pipeline.** To ensure high-quality annotations, we implement a three-stage quality control process (Figure 1 (a)). **Stage 1: Expert Inspection.** Two board-certified radiologists (each >10 years' experience) examine 2,000 reports to enumerate failure modes (*e.g.*, missing injections, nonsensical outputs). **Stage 2: Script Validation.** Automated scripts validate formatting consistency, detect unchanged or malformed edits, remove redundant symbols, and ensure that exactly one or the intended number of edits exist per sample. Guided by Stage-1 failure types, scripts scan the remaining ∼24k reports and flag ∼900 candidates for human review. **Stage 3: Final Review.** The annotators re-examine all flagged cases (∼30 s/case), resolve edge cases and ambiguities, and correct

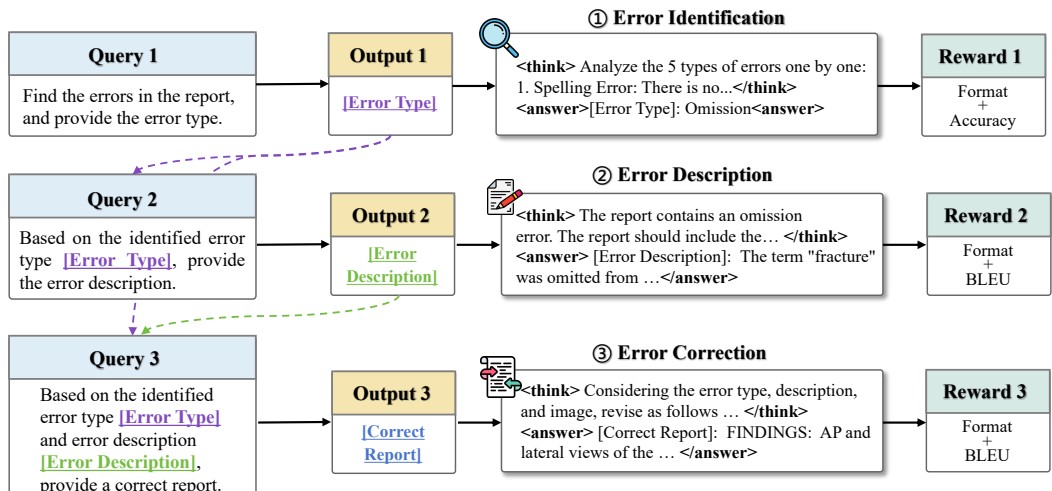

Figure 3: Illustration of our multi-step reinforcement-learning framework: the model sequentially performs error identification, description, and correction, with each stage guided by a tailored reward.

any residual inconsistencies. This combined human–in-the-loop pipeline ensures both scalability and reliability in dataset construction.

**Dataset Composition and Availability.** The CorBenchX consists of clean–corrupted report pairs with detailed annotations, including error type, span-level edits, and error descriptions. An example of a CXR image and its associated reports is shown in Figure 2, while Table 1 summarizes the error categories and their distributions. More details are reported in Appendix D. The dataset serves as a comprehensive benchmark for developing and evaluating radiology report error detection and correction systems. The complete dataset has been submitted to PhysioNet and is currently under review; it will be publicly available soon.

## 4 MULTI-STEP REINFORCEMENT LEARNING

Correcting radiology report errors requires precise localization of erroneous spans and flexible, context-aware revision strategies. Due to the diverse linguistic patterns across error types, fixed or templated supervision is often inadequate. To address this, we introduce a novel method and formulate the task as a three-stage reinforcement learning problem that promotes step-by-step reasoning and fine-grained correction. We adopt Group Relative Policy Optimization (GRPO) as the training objective to guide the model toward clinically consistent and contextually appropriate revisions.

### 4.1 THREE STAGE REINFORCEMENT LEARNING OPTIMIZATION

The report correction task can be decomposed into three stages: **error identification** $\rightarrow$ **error description** $\rightarrow$ **error correction**. Based on this, we design a multi-step approach that breaks the complete trajectory into multiple sub-trajectories to encourage the model to perform clear and targeted reasoning at each step, thereby enabling supervision over intermediate reasoning processes, as illustrated in Figure 3. Formally, the reasoning trajectory is denoted as

$$\mathcal{T} = ((Q_1, O_1), \ldots, (Q_K, O_K)), \tag{1}$$

where $Q_k$ and $O_k$ denote the model's query and output at each step, respectively. $K$ represents the total number of steps required by the reasoning trajectory, which is set to 3 in our task. The first state $Q_1$ serves as the initial prompt. Each subsequent query $Q_k$ contains the previous query $Q_{k-1}$ and the corresponding output $O_{k-1}$.

**Step 1: Error Identification.** First, we supervise the model to correctly identify the error type by optimizing classification accuracy. The reward for this step, denoted as $R_1$, is the sum of the Format Reward and the Accuracy Reward. **Format Reward:** The format reward $R_{format} \in \{0, 1\}$

is designed to ensure that the model encloses its reasoning within the designated tags (e.g., <think> and </think>) and wraps the final answer within <answer> and </answer> tags.

$$R_{format} = \mathbb{1}(match(content)), \tag{2}$$

where $match$ denotes the regular expression matching operation.

**Accuracy Reward:** The accuracy reward $R_{acc} \in \{0, 1\}$ is set to 1 if the model correctly identifies the current error type, and 0 otherwise.

$$R_{acc} = \mathbb{1}(Err_{pred} = Err_{gt}), \tag{3}$$

where $Err_{pred}$ denotes the model's predicted error type, and $Err_{gt}$ refers to the ground truth.

**Step 2: Error Description.** Based on the Step 1, we perform error description to help the model better understand and localize different types of errors. This step also enables the model to provide users with more detailed references and explanations during interaction. We supervise the quality of error description using the Format Reward and the BLEU Reward. The Format Reward is the same as above, and the **BLEU Reward** is defined as follows.

$$R_{bleu} = BLEU(Des_{pred}, Des_{gt}), \tag{4}$$

where $Des_{pred}$ denotes the model's predicted description, and $Des_{gt}$ refers to the ground truth.

**Step 3: Error Correction.** Building on the previous two steps, the model conducts evidence-based error correction, with the accuracy of the corrections supervised by the Format and BLEU Reward. Here we employ BLEU rather than clinically-grounded metrics like RadGraph-F1 or CheXbert because Stage 1 already explicitly ensures classification accuracy, a goal aligned with RadGraph-F1/CheXbert. Moreover, BLEU offers significantly higher computational efficiency, which is critical for feasible reinforcement learning training.

## 4.2 TRAINING WITH GRPO

The model's policy is optimized to maximize the cumulative reward over the entire trajectory for 3 stages RL learning, formulated as:

$$J(\theta) = \sum_{k=1}^{K} J^k(\theta). \tag{5}$$

Here, $\pi_\theta$ is the policy parametrized by $\theta$. $J^k(\theta)$ denotes the optimization objective at step $k$. We employ GRPO Guo et al. (2025), a variant of PPO Schulman et al. (2017) that introduces advantage normalization within grouped samples, as the optimization objective at each step. The objective guides the policy to generate structurally coherent and instruction-following report error corrections.

$$\begin{aligned} J^k(\theta) = \mathbb{E}[q^k \sim P(Q^k),\ \{o_i^k\}_{i=1}^{G} \sim \pi_{\theta_{old}}(O^k|q^k)] \frac{1}{G} \sum_{i=1}^{G} \bigg( \min\Big( \frac{\pi_\theta(o_i^k|q^k)}{\pi_{\theta_{old}}(o_i^k|q^k)} A_i^k, \\ \mathrm{clip}\Big( \frac{\pi_\theta(o_i^k|q^k)}{\pi_{\theta_{old}}(o_i^k|q^k)}, 1 - \varepsilon,\ 1 + \varepsilon \Big) A_i^k \Big) - \beta D_{\mathrm{KL}}(\pi_\theta \| \pi_{\mathrm{ref}}) \bigg). \end{aligned} \tag{6}$$

where $\pi_{\theta_{old}}$ presents the old policy model, $Q_k$ is the query for step $k$, $\varepsilon$ and $\beta$ are hyperparameters, G denotes the number of outputs within a group. $A_i^k$ is the advantage calculated based on rthe relative rewards of the outputs within each group. During training, the number of grouped samples is set to 8.

## 5 EVALUATION

### 5.1 EXPERIMENTAL SETTINGS

**Evaluation Models.** We evaluate nine vision-language models (VLMs) alongside our proposed method under a zero-shot setting for two tasks: error detection and error correction in chest X-ray reports. The evaluated models include six open-source VLMs: MedVLM-R1 Pan et al. (2025), CheXagent Chen et al. (2024), InternVL3-8B Zhu et al. (2025), Baichuan-Omni-1.5-7B Li et al.

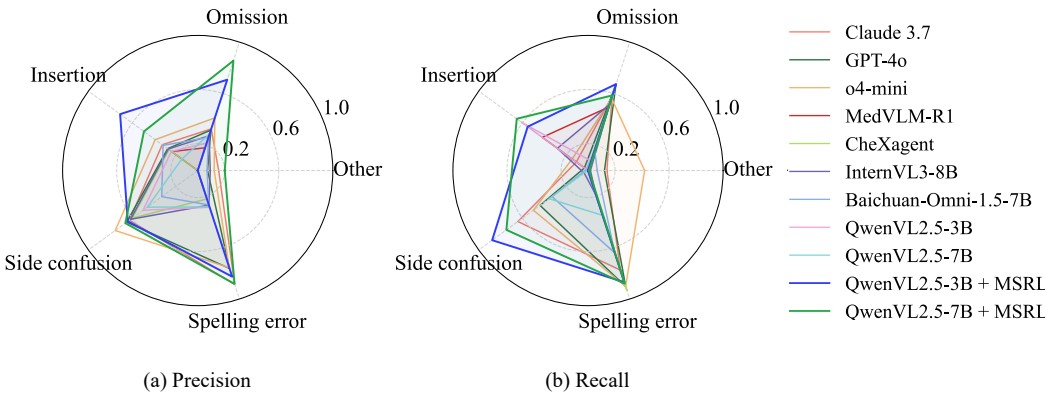

(a) Precision                    (b) Recall

Figure 4: Precision and recall for single-error detection across various VLMs and models enhanced by our MSRL, broken down by the five error categories.

Table 2: Evaluation results on single-error report correction (report-level)/( sentence-level ). The highest score in each column is highlighted in pink , and the second-best in blue .

| Model | BLEU | ROUGE | BERTScore | SembScore | CheXbertF1 | RadGraphF1 |
|-------|------|-------|-----------|-----------|-----------|-----------|
| Claude 3.7 sonnet | 0.852 | 0.914 | 0.982 | 0.817 | 0.935 | 0.889 |
| GPT-4o | 0.787 | 0.872 | 0.964 | 0.782 | 0.898 | 0.843 |
| o4-mini | 0.853 | 0.924 | 0.981 | 0.865 | 0.954 | 0.905 |
| MedVLM-R1 | 0.315 | 0.469 | 0.841 | 0.459 | 0.610 | 0.484 |
| CheXagent | 0.519 | 0.669 | 0.898 | 0.695 | 0.795 | 0.674 |
| InternVL3-8B | 0.768 | 0.848 | 0.948 | 0.777 | 0.903 | 0.813 |
| Baichuan-Omni1.5-7B | 0.792 | 0.876 | 0.966 | 0.784 | 0.899 | 0.826 |
| QwenVL2.5-3B | 0.786 | 0.892 | 0.971 | 0.807 | 0.907 | 0.863 |
| QwenVL2.5-7B | 0.830 | 0.906 | 0.974 | 0.793 | 0.905 | 0.863 |
| **QwenVL2.5-3B+MSRL** | 0.938 | 0.971 | 0.993 | 0.839 | 0.951 | 0.931 |
| **QwenVL2.5-7B+MSRL** | 0.960 | 0.984 | 0.997 | 0.905 | 0.984 | 0.958 |
| Claude 3.7 sonnet | 0.345 | 0.477 | 0.862 | 0.789 | 0.815 | 0.416 |
| GPT-4o | 0.365 | 0.550 | 0.870 | 0.795 | 0.843 | 0.465 |
| o4-mini | 0.386 | 0.547 | 0.876 | 0.852 | 0.878 | 0.482 |
| MedVLM-R1 | 0.282 | 0.441 | 0.826 | 0.508 | 0.646 | 0.406 |
| CheXagent | 0.326 | 0.481 | 0.840 | 0.665 | 0.706 | 0.413 |
| InternVL3-8B | 0.516 | 0.719 | 0.914 | 0.775 | 0.863 | 0.606 |
| Baichuan-Omni1.5-7B | 0.504 | 0.713 | 0.920 | 0.762 | 0.862 | 0.591 |
| QwenVL2.5-3B | 0.486 | 0.702 | 0.919 | 0.773 | 0.836 | 0.580 |
| QwenVL2.5-7B | 0.467 | 0.686 | 0.911 | 0.790 | 0.849 | 0.554 |
| **QwenVL2.5-3B+MSRL** | 0.481 | 0.701 | 0.921 | 0.764 | 0.824 | 0.558 |
| **QwenVL2.5-7B+MSRL** | 0.400 | 0.536 | 0.868 | 0.897 | 0.929 | 0.446 |

(2025), QwenVL2.5-3B, and QwenVL2.5-7B Bai et al. (2025); and three closed-source models: Claude 3.7 Sonnet, GPT-4o Achiam et al. (2023), and o4-mini OpenAI (2025).

**Implementation Details.** All experiments are conducted on NVIDIA A800 GPUs. For each model, we prompt it to perform two tasks: (1) identify and classify the error type in the error report, and (2) generate a corrected version of the report. No additional fine-tuning or in-domain training is performed. Detailed hyperparameters and prompting templates are provided in Appendix A.

**Evaluation Metrics.** We assess each model's performance along three dimensions: (1) Error detection: measured by precision and recall over the five error types; (2) Error correction in report level: assessed with two word level metrics: BLEU Papineni et al. (2002) and ROUGE Lin (2004), two semantic level metrics: BERTScore Zhang et al. (2019) and SembScore Smit et al. (2020), and two clinical efficacy level metrics: CheXbert Smit et al. (2020) and RadGraph-F1 Yu et al. (2023b);

and (3) Error correction in sentence level: apply the same suite of six metrics to the individual corrected sentences, enabling fine-grained assessment of local edits.

## 5.2 EXPERIMENTAL RESULTS AND ANALYSIS

We first evaluate the performance of nine baseline VLMs on both single-error and multi-error detection and correction tasks. We then compare these results with our proposed MSRL-enhanced models—**QwenVL2.5-3B+MSRL** and **QwenVL2.5-7B+MSRL**—to assess the effectiveness of multi-step reinforcement learning in improving fine-grained clinical reasoning and radiology report correction. Finally, we conduct an ablation study to validate the contribution of our multi-step RL framework compared to standard single-step reinforcement learning.

**Results on Single-error Detection and Correction.** Figures 4 (a) and 4 (b) present per-error-type precision and recall for single-error detection across nine evaluated vision-language models (VLMs). Table 2 summarizes the corresponding error correction performance, evaluated using six metrics at both the report (upper part) and sentence levels (lower part). As shown in Figure 4, **o4-mini** achieves the best overall detection performance, with an average precision of 0.486 and recall of 0.506. In terms of correction quality (Table 2), closed-source models—Claude 3.7 Sonnet, GPT-4o, and o4-mini—consistently outperform their open-source counterparts in report-level metrics, with o4-mini ranking highest across all evaluation scores. Within open-source models, **QwenVL2.5-7B** leads the pack, whereas MedVLM-R1 performs markedly worse. Generally, sentence-level metrics (lower part) are substantially lower than report-level scores, demonstrating that localized, span-level evaluation reveals challenges masked by full-report metrics. Across all models, current error correction capabilities of existing VLMs fall short of clinical-grade reliability, reinforcing the need for more targeted and interpretable strategies.

In Table 2, QwenVL2.5-7B+MSRL exhibits a modest reduction on purely lexical metrics (e.g., BLEU, ROUGE) for sentence-level evaluation, yet it demonstrates significant improvements on semantic and clinical entity consistency metrics such as SembScore and CheXbert F1. This reflects a key distinction between surface-level textual similarity and clinically meaningful correctness. In real-world radiology workflows, preserving diagnostic content, maintaining entity consistency, and preventing clinically harmful alterations are far more important than matching reference sentences token-by-token. Notably, this phenomenon is not unique to our model; strong closed-source baselines such as GPT-4o exhibit the same trend, performing modestly on lexical metrics while excelling on semantic and clinical measures. Overall, these results indicate that MSRL prioritizes clinically faithful corrections over superficial lexical overlap, which better aligns with the requirements of practical radiology reporting.

**Results on Multi-error Detection and Correction.** Figure 5 (a) and Figure 5 (b) depict per-error-type precision and recall for multi-error detection across all evaluated VLMs. Closed-source models again dominate: Claude 3.7 achieves the highest average precision (0.612), while o4-mini attains the highest average recall (0.580), both substantially outperforming open-source models. Table 3 reports multi-error correction performance under the same six metrics. At the report level, QwenVL2.5-3B is the top open-source performer. The results are far lower than those for single-error correction, underscoring the substantial challenge that multi-error correction poses for current VLMs. At the sentence level, Baichuan-Omni1.5-7B obtains the best results. Notably, o4-mini underperforms because it paraphrases entire reports instead of making focused span-level corrections.

**Effectiveness of Multi-step Reinforcement Learning.** We perform our MSRL on Qwen-2.5-VL 3B and Qwen-2.5-VL 7B and compare its performance with other VLMs. As shown in Figures 4 and Figures 5, our method achieves an average increase of 38.3% in precision and 30.5% in recall on the single error detection task with Qwen-2.5-VL-7B. Similarly, for multi-error detection, we observe an average improvement of 23.6% in precision and 1.5% in recall, validating the generalization capability of the model. Notably, when our model is initialized with Qwen-2.5-VL 3B, its classification accuracy on the "other" category remains at a very low level. The underlying reason is that Qwen-2.5-VL 3B, under zero-shot settings, fails to recognize the "other" category and tends to ignore its analysis during the reasoning process (the content within the <think> </think>). This observation highlights that without early-stage instruction fine-tuning, RL alone yields suboptimal reasoning performance, which has been approved in Guo et al. (2025); Liu et al. (2025). For report-level correction, Table 2 shows that QwenVL2.5-3B and 7B models trained with multi-step RL outperform their zero-shot

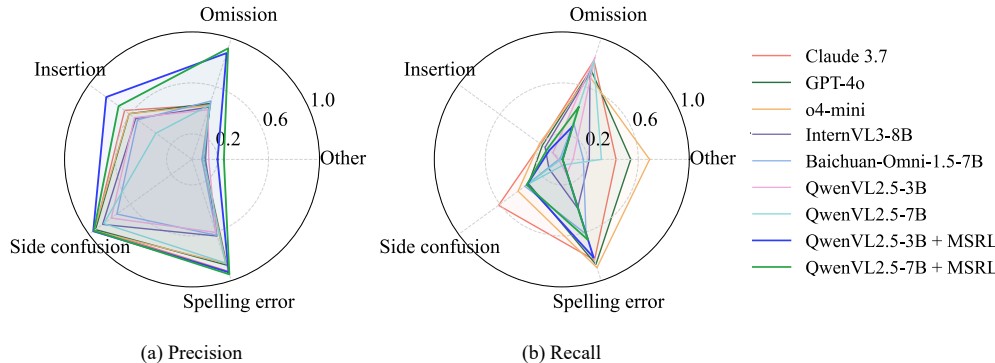

Figure 5: Precision and recall for multi-error detection across various VLMs and models enhanced by our MSRL, broken down by the five error categories.

Table 3: Evaluation results on multi-error report correction (report-level)/( sentence-level ). The highest score in each column is highlighted in pink , and the second-best in blue .

| Model | BLEU | ROUGE | BERTScore | SembScore | CheXbertF1 | RadGraphF1 |
|---|---|---|---|---|---|---|
| Claude 3.7 sonnet | 0.701 | 0.817 | 0.959 | 0.724 | 0.856 | 0.773 |
| GPT-4o | 0.629 | 0.769 | 0.935 | 0.669 | 0.801 | 0.719 |
| o4-mini | 0.404 | 0.619 | 0.909 | 0.670 | 0.788 | 0.596 |
| InternVL3-8B | 0.685 | 0.808 | 0.940 | 0.682 | 0.811 | 0.748 |
| Baichuan-Omni1.5-7B | 0.755 | 0.875 | 0.966 | 0.753 | 0.876 | 0.817 |
| QwenVL2.5-3B | 0.742 | 0.859 | 0.964 | 0.736 | 0.855 | 0.805 |
| QwenVL2.5-7B | 0.728 | 0.847 | 0.959 | 0.712 | 0.833 | 0.780 |
| **QwenVL2.5-3B+MSRL** | 0.874 | 0.940 | 0.985 | 0.794 | 0.908 | 0.866 |
| **QwenVL2.5-7B+MSRL** | 0.900 | 0.958 | 0.992 | 0.852 | 0.948 | 0.898 |
| Claude 3.7 sonnet | 0.461 | 0.666 | 0.919 | 0.684 | 0.781 | 0.577 |
| GPT-4o | 0.502 | 0.735 | 0.925 | 0.712 | 0.807 | 0.593 |
| o4-mini | 0.297 | 0.575 | 0.893 | 0.724 | 0.813 | 0.505 |
| InternVL3-8B | 0.538 | 0.761 | 0.932 | 0.697 | 0.812 | 0.631 |
| Baichuan-Omni1.5-7B | 0.591 | 0.810 | 0.952 | 0.737 | 0.828 | 0.680 |
| QwenVL2.5-3B | 0.569 | 0.783 | 0.943 | 0.709 | 0.810 | 0.657 |
| QwenVL2.5-7B | 0.560 | 0.800 | 0.945 | 0.714 | 0.806 | 0.640 |
| **QwenVL2.5-3B+MSRL** | 0.647 | 0.848 | 0.967 | 0.753 | 0.848 | 0.693 |
| **QwenVL2.5-7B+MSRL** | 0.636 | 0.827 | 0.961 | 0.829 | 0.901 | 0.691 |

baselines by 7.4% and 5.2% on single-error correction. Sentence-level gains are even larger. On the more challenging multi-error task (Table 3), our model improvements reach 6.8% and 11.5%, highlighting the effectiveness and generalization ability of our MSRL.

**Ablation Studies.** As shown in Table 4, we compare our MSRL with single-step RL, which incorporates all processes into a single inference and simultaneously optimizes the Accuracy Reward, Format Reward, and BLEU Reward. This approach fails to effectively follow instructions step by step, resulting in an average performance gap of 13.3% compared to MSRL.

Table 4: Ablation studies on RL and MSRL for single-error correction.

| Model | Method | BLEU | ROUGE | BERTScore | SembScore | CheXbertF1 | RadGraphF1 |
|---|---|---|---|---|---|---|---|
| QwenVL2.5-3B | RL | 0.788 | 0.882 | 0.944 | 0.798 | 0.916 | 0.853 |
| | MSRL | 0.938 | 0.971 | 0.993 | 0.839 | 0.951 | 0.931 |
| QwenVL2.5-7B | RL | 0.873 | 0.939 | 0.978 | 0.838 | 0.945 | 0.906 |
| | MSRL | 0.960 | 0.984 | 0.997 | 0.905 | 0.984 | 0.958 |

Table 5: OOD evaluation results on IU-Xray dataset for single-error correction.

| Model | BLEU | ROUGE | BERTScore | SembScore | CheXbert F1 | RadGraph F1 |
|-------|------|-------|-----------|-----------|-------------|-------------|
| QwenVL2.5-3B | 0.338 | 0.411 | 0.754 | 0.435 | 0.400 | 0.399 |
| + MSRL | **0.829** | **0.958** | **0.974** | **0.641** | **0.830** | **0.921** |
| QwenVL2.5-7B | 0.074 | 0.091 | 0.624 | 0.377 | 0.129 | 0.088 |
| + MSRL | **0.840** | **0.964** | **0.975** | **0.713** | **0.958** | **0.935** |

**Out-of-Distribution (OOD) Evaluation.** To assess the robustness of the MSRL, we conduct evaluation on the IU-Xray corpus. We uniformly sample 600 reports, inject synthetic errors using the same taxonomy, and then evaluate zero-shot correction performance. As shown in Table 5, augmenting QwenVL2.5 with MSRL yields large, consistent gains across lexical, semantic, and clinical entity metrics, for both 3B and 7B backbones.

# 6 CONCLUSION

We present CorBenchX, the first large-scale benchmark for automated error detection and correction in chest X-ray reports. By synthesizing 26,326 clinically motivated error cases via DeepSeek-R1, we enable a rigorous evaluation of both open- and closed-source LLMs. Our experiments reveal that even the best model achieves just 50.6 % error-type detection accuracy and remain below clinical-grade correction. We further propose MSRL that sequentially supervises error identification, description, and correction, yielding substantial gains over the baselines.

**Limitations & Future Work.** CorBenchX currently targets chest X-ray reports and does not model errors tied to prior imaging or patient history. We will extend to CT/MRI and integrate EHR context to assess longitudinal, patient-specific error detection and correction.

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

# Appendix

## A  IMPLEMENTATION DETAILS

### A.1  HYPERPARAMETERS FOR VLM EVALUATION

For all VLMs evaluated, the image resolution for vision input is $336 \times 336$, and the maximum number of generated tokens is 1024. The experiments of all publicly available models were conducted on a single NVIDIA RTX 3090 GPU, and the proprietary models were evaluated with official APIs.

We used Claude 3.7 sonnet of `20250219` version, GPT-4o of `gpt-4o-2024-11-20` version, and o4-mini of `o4-mini-2025-04-16` version.

### A.2  EXPERIMENTAL SETTINGS ON MULTI-STEP REINFORCEMENT LEARNING

The training and test splits are detailed in Table 6.

Table 6: Training and test split.

| Error type | Training | Test (single error) | Test (multi error) |
|---|---|---|---|
| Omission | 4,273 | 1,061 | 933 |
| Insertion | 4,380 | 1,280 | 1,275 |
| Spelling Error | 4,611 | 1,160 | 1,348 |
| Side Confusion | 4,540 | 1,482 | 1,593 |
| Other | 1,080 | 279 | 153 |
| Total | 18,884 | 5,262 | 5,302 |

The hyperparameter configurations for our MSRL are listed in Table 7.

Table 7: Detailed training hyperparameters for our MSRL.

| Configuration | MSRL |
|---|---|
| Model Init | Qwen2.5-VL |
| Global batch size | 128 |
| Learning rate | $2 \times 10^{-5}$ |
| Weight decay | 0 |
| Resolution | 336 |
| Num Generations | 8 |
| Optimizer | AdamW |
| Epochs | 2 |
| GPU Usage | 8 NVIDIA A800 |
| Training time | 3B-43h; 7B-51h |

## B  PROMPTS

In this section, we provide the prompts for report error correction and error report generation.

### B.1  PROMPTS FOR REPORT ERROR CORRECTION

In this section, we provide the precise prompt templates employed for both single-error and multi-error correction under zero-shot and MSRL evaluation. The prompt for single-error correction is given in Sec. B.1.1 and the prompt for multi-error correction is given in Sec. B.1.2.

### B.1.1 SINGLE ERROR CORRECTION

---

**Single Error Correction Prompt**

Report:
FINDINGS: PA and lateral views of the chest are provided. Elevated right hemidiaphragm is unchanged. There is minimal plate-like right basal atelectasis. No focal consolidation, effusion, pneumothorax. The cardiomediastinal silhouette appears stable. Bony structures are intact.
IMPRESSION: No acute findings in the chest.

You are a senior clinician reviewing a diagnostic report. The report may inadvertently contain common errors in the following 5 categories:
1. Omission: The omission of relevant words or expressions, including deletions or missing words (e.g., "fracture" instead of "no fracture").
2. Insertion: The unintentional insertion of incorrect words or expressions, including inappropriate words, wrong word substitutions, or extra words (e.g., "abnormal" instead of "normal").
3. Spelling Error: Spelling mistakes or word truncations due to manual text processing errors (e.g., "pnuemothorax" instead of "pneumothorax").
4. Side Confusion: Errors involving laterality or orientation (e.g., "right" instead of "left," or "lateral" instead of "medial").
5. Other: Includes mistakes in units of measurement (e.g., "centimeter" vs "millimeter") or punctuation mistakes.

Your task is to detect any errors present in the report and correct them.

Output Format:
Please only output content strictly according to the format below and there is only one error, do not output multiple errors, do not output other content, the format is:
[Error Type]: (Omission / Insertion / Spelling Error / Side Confusion / Other), your should strictly follow the format.
[Error Description]: [A concise explanation of the error]
[Correct Report]: [Based on the detected errors, revise the original report and output the corrected version of the report.]

Ensure that all errors detected are clearly described and the output strictly follows the structure and format provided above.

---

### B.1.2   MULTI-ERROR CORRECTION

---

**Multi-Error Correction Prompt**

Report:
FINDINGS: Cardiac silhouette size remains moderately enlarged due
to prominent epicardial fat pads.  Mediastinal contour is unchanged,
and stably widened compatible with mediastinal lipomatosis.  Hilar
contours are normal.  Lungs are clear and the pulmonary vascularity
is normal.  Pleural thickening is noted unilaterally due to pleural
fat deposition.  No pleural effusion or pneumothorax is seen.
There are no acute osseous abnormalities.
IMPRESSION: No acute cardiopulmonnary process.

You are a senior clinician reviewing a diagnostic report.  The
report may inadvertently contain common errors in the following 5
categories:
1.  Omission:  The omission of relevant words or expressions,
including deletions or missing words (e.g., "fracture" instead of
"no fracture").
2.  Insertion:  The unintentional insertion of incorrect words
or expressions, including inappropriate words, wrong word
substitutions, or extra words (e.g., "abnormal" instead of
"normal").
3.  Spelling Error:  Spelling mistakes or word truncations due
to manual text processing errors (e.g., "pnuemothorax" instead of
"pneumothorax").
4.  Side Confusion:  Errors involving laterality or orientation
(e.g., "right" instead of "left," or "lateral" instead of
"medial").
5.  Other:  Includes mistakes in units of measurement (e.g.,
"centimeter" vs "millimeter") or punctuation mistakes.

Your task is to detect any errors present in the report and correct
them.

Output Format:
Please only output content strictly according to the format below
and there may exist multiple errors.  Do not output other content,
the format is:
[Error Type]:  (Omission / Insertion / Spelling Error / Side
Confusion / Other), your should strictly follow the format.
[Error Description]:  [A concise explanation of the error]
[Correct Report]:  [Based on the detected errors, revise the
original report and output the corrected version of the report.]

Ensure that all errors detected are clearly described and the
output strictly follows the structure and format provided above.

---

## B.2 PROMPTS FOR ERROR REPORT GENERATION

In this section, we provide the exact prompts used to synthesize our error-injected chest X-ray reports. The prompt for single-error injection is given in Sec. B.2.1, and the prompt for multi-error injection appears in Sec. B.2.2.

### B.2.1 SINGLE ERROR INJECTION

Below is the prompt used to inject a single Omission error into each report. To generate reports with any of the other error categories, replace the "Omission" instruction and its description with the desired error type and corresponding explanation.

---

**Single Error Injection Prompt – Omission Error**

```
Report:
FINDINGS: PA and lateral views of the chest are provided.  Elevated
right hemidiaphragm is unchanged.  There is minimal plate-like left
basal atelectasis.  No focal consolidation, effusion, pneumothorax.
The cardiomediastinal silhouette appears stable.  Bony structures
are intact.
IMPRESSION: No acute findings in the chest.

Add exactly one Omission error into the above report.  An
Omission error is defined as the omission of relevant words or
expressions, which encompasses both deletions and missing words
(e.g., "fracture" instead of "no fracture").
```

**Output Format:**
```
First, output the modified report with one error introduced.  After
the report, clearly identify and explain the introduced errors in
the following format:
[Original Text]:  "XXX"
[Revised Text]:  "YYY"
[Error Type]:  Omission
[Error Description]:  e.g.,' Omission/Missing of an expression
in the FINDINGS section', or 'Omission of "XXX" in the FINDINGS
section'.  Do not use "changed", "modified", "revised", or
"original report" in the Error Description.
```

**Ensure that:**
```
- Only one error is introduced per report.
- The output remains medically realistic.
- The formatting is consistent and follows the structure exactly as
specified.
```

---

### B.2.2 MULTI-ERROR INJECTION

Below is the prompt used to inject three errors per report. To generate two-error variants, simply replace "three" with "two" in the task instructions.

---

**Multi-Error Injection Prompt**

```
Report:
FINDINGS: PA and lateral views of the chest are provided.  Elevated
right hemidiaphragm is unchanged.  There is minimal plate-like left
basal atelectasis.  No focal consolidation, effusion, pneumothorax.
The cardiomediastinal silhouette appears stable.  Bony structures
are intact.
IMPRESSION: No acute findings in the chest.

You are a junior clinician reviewing the above diagnostic report.
As a junior clinician, you may inadvertently introduce some common
errors into the report.  Your task is to introduce three errors
into the report.  The error should be randomly selected from the
following five categories:
1.  Omission:  The omission of relevant words or expressions, which
encompasses both deletions and missing words (e.g., "fracture"
instead of "no fracture").
2.  Insertion:  The unintentional insertion of incorrect words
or expressions, including inappropriate words, incorrect word
substitutions, insertions, or word confusions (e.g., "abnormal"
instead of "normal").
3.  Spelling Errors:  Spelling mistakes, including word truncations,
likely due to manual text processing by radiologists through typing
errors or inaccurate selection of text that is to be removed or
edited, avoid change pneumothorax to pnuemothorax.
4.  Side Confusion:  Errors involving side or orientation (e.g.,
"right" instead of "left," "lateral" instead of "medial").
5.  Other Errors:  Including mistakes in units of measurement (e.g.,
"centimeter" vs "millimeter"), and punctuation mistakes.

Output Format:
First, output the modified report with three errors introduced.
After the report, clearly identify and explain the introduced
errors in the following format:
[Original Text]:  "XXX"
[Revised Text]:  "YYY"
[Error Type]:  (Omission / Insertion / Spelling Error / Side
Confusion / Other)
[Error Description]:  e.g., Omission of "XXX" in the FINDINGS
section, misspelling XXX as XXX, or insertion of XXX. Do not use
"changed", "modified", "revised", or "original report" in the Error
Description.

Ensure that:
- Three errors are introduced per report.
- The output remains medically realistic.
- The formatting is consistent and follows the structure exactly as
specified.
```

---

## C LICENSES OF PUBLIC DATASET

The MIMIC-CXR v2.0.0 dataset, from which CorBenchX is derived, is released under the PhysioNet Credentialed Health Data License 1.5.0, which requires all users to register for a PhysioNet account, complete human-subjects protection training, and sign a Data Use Agreement (DUA) prohibiting any attempt to re-identify patients or share the raw data.

# D  DETAILS ABOUT THE CURATED DATASET

Table 8: Error categories in CorBenchX: description, clinical rationale, and representative examples.

| Error Type | Description & Clinical Rationale | Representative Examples |
|---|---|---|
| Omission | Removal of clinically significant modifiers, negations, or key terms that change finding presence or certainty. Tests the model's ability to detect missing critical information. | "Osseous structures are grossly intact." → "Osseous structures are intact." "No pleural effusion." → "Pleural effusion." "Mild vascular congestion." → "Vascular congestion." |
| Insertion | Addition of one or two words that alter diagnostic meaning or certainty, evaluating sensitivity to introduced false findings. | "No pneumonia." → "Mild pneumonia." "Heart size is normal." → "Heart size is abnormal." |
| Spelling error | Single-word misspellings or typographical errors of clinically important terms; tests robustness to common dictation/typing mistakes. | "pneumothorax" → "pnuemothorax"; "basilar" → "basillar"; "esophagus" → "esophogus"; "edema" → "edemma". |
| Side confusion | Incorrect laterality or orientation, e.g., swapping left/right or mislabeling lobes; directly affects interventions. | "left" ↔ "right"; "bilateral" → "right" (or "left"); "right upper lobe" → "right lower lobe". |
| Other | Miscellaneous errors such as incorrect numeric measurements/units or punctuation changes that alter quantitative interpretation or report structure. | Measurement: "1.5 mm" → "1.5 cm" . Punctuation: missing/extra periods or commas that change sentence parsing. |

Table 9: Mean number of words modified per injected error across error types.

| Error Type | Omission | Insertion | Spelling Error | Side Confusion | Other |
|---|---|---|---|---|---|
| Mean # of Modified Words | 1 | 1–2 | 1–2 | 1–4 | 1–2 |

