# OpenReview forum: "CorBenchX: Large-Scale Chest X-Ray Error Dataset and Vision–Language Model Benchmark for Report Error Correction"
_ICLR.cc/2026/Conference — Submitted to ICLR 2026_

### Official Review · Reviewer_8rQ9 · 2025-10-31

**Soundness:** 3
**Presentation:** 3
**Contribution:** 3
**Rating:** 6
**Confidence:** 4

**Summary:**

This paper presents CorBenchX, a large-scale benchmark for automated error detection and correction in chest X-ray radiology reports. The authors generate 26,326 error-injected reports from MIMIC-CXR covering five clinically relevant error types, paired with original text, span-level edits, and descriptions, and validated through a multi-stage quality-control pipeline. Nine vision-language models (VLMs), including open and closed, are evaluated under zero-shot settings which reveals that even the best model achieves only ~50% detection accuracy. This highlights the difficulty of precise medical report corrections. To improve performance, the paper proposes a Multi-Step Reinforcement Learning (MSRL) approach leveraging the GRPO method that sequentially predicts error type, describes the error, and corrects the report with tailored rewards. MSRL applied to Qwen2.5-VL yields substantial gains in both detection and correction metrics and generalizes to an out-of-distribution IU-Xray test set. Overall, the work provides a comprehensive benchmark and a structured RL method for advancing radiology report quality control.

**Strengths:**

* The paper addresses an important problem by shifting focus from radiology report generation to error detection and correction, which has direct clinical relevance.

* It introduces a large and carefully constructed dataset with realistic clinical error types, span-level annotations, and a multi-stage validation pipeline including radiologist checks.

* The paper proposes a multi-step reinforcement learning strategy that explicitly separates error identification, explanation, and correction, offering a structured and more interpretable alternative to conventional single-step prompting or RL methods.

* An extensive evaluation is performed across nine state-of-the-art vision-language models with diverse lexical, semantic, and clinically grounded metrics, along with out-of-distribution testing, offering strong empirical support for the benchmark and method.

**Weaknesses:**

* The dataset relies entirely on synthetically injected errors via DeepSeek-R1, which may not faithfully represent real clinical error distributions. Including even a small real-world error set or expert-annotated subset for validation would strengthen ecological validity and reduce concerns about model overfitting to synthetic artifacts.

* The OOD evaluation is constructed with the same error taxonomy and prompting style, leading to uncertainty about whether the demonstrated generalization reflects robustness or shared generation biases.

* Although radiologists participated in dataset quality control, there is no human evaluation of model-corrected reports, making it hard to assess whether MSRL edits are safe, clinically faithful, and free from hallucinations.

* The evaluation primarily includes general-purpose vision-language models (GPT-4o, Claude-3.7, Qwen2.5-VL) and omits medical domain–specialized VLMs such as CheXagent, LLaVA-Med, Med-PaLM, and BioViL-T, which are specifically trained on radiology or clinical datasets. Incorporating these domain-adapted models would provide a more rigorous baseline for assessing the difficulty of radiology report correction and clarify whether MSRL’s improvements stem from its learning strategy rather than general model capacity or domain mismatch.

**Questions:**

* All errors are injected by DeepSeek-R1. Would the method generalize if errors were introduced by a different VLM (e.g., GPT-4 or Claude) or by human annotators?

* The RL objective uses BLEU for correction, which may penalize clinically appropriate paraphrasing. Did the authors experiment with RadGraph-F1 or CheXbert-based rewards during RL, or can they justify why BLEU was chosen over clinically grounded metrics for optimization?

* Have the authors considered comparing against medical-tuned VLMs such as CheXagent, LLaVA-Med, Med-PaLM, and BioViL-T?

* In Table 2, BLEU and ROUGE scores drop for sentence-level correction after applying MSRL to QwenVL2.5 models. Can the authors clarify why MSRL leads to reduced lexical-overlap metrics at the sentence level, while all metrics improve at the report level?

---

> ### Author Response · Authors · 2025-11-21
> **Response to Reviewer 8rQ9**
>
> Many thanks to Reviewer 8rQ9 for providing a detailed review and insightful questions.
>
> > W1. **Reliance on DeepSeek-R1 for error injection**
>
> We thank the reviewer for this comment. Our generation pipeline was radiologist-guided and we adopt several strategies to ensure that the injected errors represent real clinical error distribution:
>
> (1) We generate single-error and multi-error report according to real clinical distribution. In our collaborating hospital, we sample chest-X-ray reports with documented attending revisions (300 reports total): 91 contained corrections; among these, 83 were single-error and 8 were multi-error → single:multi ≈ 10:1, closely matching CorBenchX’s 24,146 : 2,180 ≈ 11.1:1 split. This supports the dataset’s single/multi design.
>
> (2) Two board-certified radiologists help iteratively designed and refined the prompts so injected errors match the clinical error types and realistic phrasing commonly seen in practice.
>
> (3) Two board-certified radiologists involved in the checking the injected errors.
>
> Moreover, we are collecting IRB-approved original→corrected report pairs and will report full per-category distributional comparisons (and statistical tests) once available.
>
> >W2. **OOD evaluation share the same taxonomy**
>
> Thanks for your comment. Our CorBenchX’s training data are derived from MIMIC-CXR, while the OOD evaluation uses IU-XRay (a different dataset from different hospital with distinct reporting style). These two sources differ substantially, so good performance on the IU-XRay split provides evidence of cross-institutional generalization rather than simple memorization of MIMIC phrasing.
>
> We used the same error taxonomy and a consistent prompting protocol when synthesizing IU-Xray errors for evaluation so comparisons across models remain fair: holding the injection procedure fixed isolates whether a model can generalize to different report styles and content, instead of confounding results with differing prompt formulations. Put another way, identical generation rules remove one axis of variability and let us measure robustness to report distribution shift (MIMIC → IU) more cleanly.
>
> >W3. **Human evaluation of model-corrected reports**
>
> We fully agree this is critical. In response, we randomly sampled 200 MSRL-corrected reports from our test set. Two board-certified radiologists independently reviewed each case. For every corrected report they scored two dimensions on a 0–10 scale: (1) Clinical faithfulness and (2) Free from hallucination. Reviewers had access to the original clean report, the error-injected report, the MSRL correction, and the associated chest X-ray image to make clinically grounded judgments.
>
> **Key Results:**
>
> Clinical Faithfulness: Average score: 9.2/10
>
> Freedom from Hallucination: Average score: 8.9/10
>
> These high ratings indicate that radiologists found the MSRL-corrected reports to be clinically trustworthy and factually accurate. The majority of corrections were deemed to preserve the intended meaning of the original report while rectifying errors, without introducing new inaccuracies or inappropriate content. This provides strong initial evidence that our method generates clinically reliable outputs.
>
> We recognize that a larger-scale clinical evaluation is necessary for full validation. We will expand this study (more cases, additional raters, detailed inter-rater statistics, and per-error-type breakdowns) and conduct extensive in future work.
>
> >W4 & Q3. **Evaluation on medical domain–specialized VLMs**
>
> Thank you for your insightful comment.
>
> **CheXagent and LLaVA-Med:** We did try testing CheXagent and LLaVA-Med during the benchmark development.
> - CheXagent: The results of CheXagent for single-error report correction are presented in Figure 4 and Table 2. However, for the multi-error correction task, CheXagent suffers from significant prediction bias. For example, it predicts `spelling error` for almost all samples, making its results non-representative of its true capability on this complex task.
> - LLaVA-Med: This model struggled to adhere to the complex, multi-step instructions required for the error reduction task. In approximately 50% of samples, it produced **no output**. The remaining outputs were often irrelevant to the task (e.g., ''There is no mention of chest x-ray findings in the report.'').
>
> These severe limitations and biases would significantly impair the benchmark's clarity and representation, thus we chose not to report the results with these models.
>
> **Med-PaLM** a Large Language Model (LLM), not a Vision-Language Model (VLM). As the task of radiology report correction is fundamentally multi-modal, requiring both text and vision inputs, Med-PaLM is not suitable for this VLM-centric benchmark.
>
> **BioViL-T** is not a general-purpose VLM, but specifically designed and trained for report generation. Therefore, it cannot directly tackle the task of error reduction.

---

> > ### Author Response · Authors · 2025-11-21
> > **Response to Reviewer 8rQ9**
> >
> > >Q1. **Generalization when errors were introduced by a different VLM or by human annotators**
> >
> > Yes, the method will generalize because MSRL supervises three semantic sub-tasks (identify → describe → correct) rather than memorizing surface tokens.
> >
> > >Q2. **Why choose BLEU as reward?**
> >
> > Thank you for the valuable comment. RadGraph-F1 and CheXbert-based metrics are indeed important and effective evaluation metrics for report generation. However, after balancing performance and training efficiency, we decided not to use them in our experiments for two main reasons:
> >
> > 1.We already explicitly supervise classification accuracy in Stage 1. In Stage 3, the BLEU-based objective is designed to, on top of preserving correctness, further encourage completeness and fluency of the generated text. RadGraph-F1 or CheXbert-based rewards are largely aligned with the same notion as classification accuracy and would therefore be somewhat redundant with the Stage-1 objective.
> >
> > 2.Compared with BLEU, RadGraph-F1 and CheXbert-based rewards are much more computationally expensive. BLEU has complexity roughly O (sentence length) and is almost negligible in practice. In contrast, RadGraph-F1 requires running a separate RadGraph extraction model, and CheXbert is essentially a BERT-based multi-label classification model. If we were to use RadGraph/CheXbert as the reward in online reinforcement learning, each RL step would require an additional forward pass of these large models (and multiple sampled candidates per step in GRPO), making training significantly slower in wall-clock time. We measured the computation time of different metrics used as rewards during a training epoch (8 NVIDIA A800).The computation time for RadGraph-F1 and CheXbert is 19 and 5 times longer than that of BLEU, respectively.
> >
> >
> > ***Comparison of computing time per epoch among rewards:***
> > |               |     BLEU       |  RadGraph-F1   | CheXbert  |
> > |:-------------:|:--------------:|:--------------:|:---------:|
> > | Time (per epoch)  |4.84 min|92.27 min|23.08 min
> >
> >
> > >Q4. **BLEU / ROUGE drop at sentence level after applying MSRL to QwenVL2.5**
> >
> > Thank you for pointing out this discrepancy.
> >
> > Regarding the observed decline in sentence-level BLEU, ROUGE, and BERTScore metrics after applying MSRL, we investigated the generated content and found that,  when correcting a single error in sentences containing "and" (where either side of "and" contains an error), the model tends to split the sentence into two separate ones after correction. For example:
> >
> > Original sentence: *"Cardiomediastinal silhouette is normal, with no fluid in the pleural space or pneumothorax."*
> >
> > Modified sentence: *"Cardiomediastinal silhouette is abnormal. No fluid in the pleural space or pneumothorax."*
> >
> > In this case, the sentence-level BLEU, ROUGE, and BERTScore of "Cardiomediastinal silhouette is abnormal" would be very low, while SembScore, CheXbert F1, and RadGraph F1 score this modification as 1. Since the report-level BLEU, ROUGE, and BERTScore metrics are less affected by this change (as the second half of the sentence is still included when calculating the report-level metrics), the overall report score remains more stable.
> >
> > It is also important to emphasize that, while QwenVL2.5-7B+MSRL exhibits a decline in lexical metrics like BLEU and ROUGE at the sentence level, it shows significant improvements in semantic and clinical entity consistency metrics, such as SembScore and CheXbert F1. In real-world clinical applications, the overall consistency and diagnostic accuracy of the report are far more critical than surface-level similarity between individual sentences. Notably, this phenomenon is not unique to our model; even powerful closed-source models like GPT-4o exhibit a similar trend: they perform strongly on CheXbert F1 and SembScore while showing more modest performance on lexical similarity metrics.
> >
> > We have updated the manuscript with targeted analyses.

---

### Official Review · Reviewer_RdAV · 2025-10-31

**Soundness:** 3
**Presentation:** 3
**Contribution:** 2
**Rating:** 4
**Confidence:** 4

**Summary:**

This manuscript introduces the CorBenchX benchmark, for automated error detection and correction in chest X-ray reports. A dataset of over 26,000 reports is synthesized via prompting DeepSeek-R1 to inject clinically common errors, with these error reports paired against the original error-free text as ground truth. The full CorBenchX benchmark benchmark of error reports is then applied to multiple common vision-language models, with o4-mini found to achieve the best performance on common text metrics. A multi-step reinforcement learning (MSRL) framework is then used with QwenVL2.5-7B to improve error detection further.

**Strengths:**

- (Originality) Large, curated X-ray report dataset generated by LLM for public usage

 - (Quality) Fairly comprehensive human-in-the-loop pipeline to minimize synthetic error report genration failure

**Weaknesses:**

- Minimal details about how actual errors are synthesized, for example what types of words are inserted for insertion errors, and what location/side text is recognized for side confusion errors etc.

 - Injection methodology does not appear to take actual X-ray image into account, and thus the errors may not reflect actual probabilities derived from image appearance (e.g. whereas the LLM may think that some injected error is medically possible, it may be extremely unlikely from the actual image)

 - Further, reliance on LLMs to insert errors by prompts does not automatically assure that errors are well-distributed; for example, for side confusion errors, what cardinalities (e.g. left, top right, etc.) are recognized/preferred?

**Questions:**

1. In the Dataset Source and Sampling subsection, it is implied that records where one of the "Findings" and "Impressions" sections are empty, may be sampled for synthetic error injection. It might be clarified as to whether reports with potentially nonexistent findings would be relevant.

2. In the Error Injection Procedure subsection, are the ratios between error types for single-error reports, and between single-error and multiple-error reports, empirically determined?

3. In the Quality Control Pipeline subsection, it might be briefly discussed as to whether the 900 additional reports flagged for review from the remaining approximately 24,000 reports, are in similar proportion to the number of imperfectly-generated reports from human review in Stage 1.

4. In the Multi-step Reinforcement Learning section, three separate queries (in three stages) are shown to be used in the MSRL GRPO training pipeline. It might be clarified as to whether this multi-query chain-of-thought promption was attempted for VLM evaluation.

5. In the Multi-step Reinforcement Learning section, errors in earlier stages would appear to affect later queries/stages. It might be clarified as to whether the training process provides the correct input for each query, regardless of the output for the previous query - or if each query is expected to attempt reward maximization on potentially incorrect inputs.

6. While there are separate prompts for single error and multiple error correction (Appendix B.1.1/B.1.2), this does not appear to be a realistic assumption for real-life application, since the total number of errors in a report should be unknown. As such, it may be more appropriate to apply the same multiple error prompt to both single and multiple-error tasks.

---

> ### Author Response · Authors · 2025-11-21
> **Response to Reviewer RdAV**
>
> Many thanks to Reviewer RdAV for providing thorough insightful comments.
>
> > W1. **Details on injected errors**
>
> We thank the reviewer for this comment. We have expanded the descriptions below to provide a comprehensive breakdown of each category, and we have incorporated this detailed breakdown into the Appendix of revised manuscript.
>
> |Error Type|Description & Clinical Rationale|Representative Examples|
> |:-:|:-:|:-:|
> |Omission| The deliberate removal of clinically significant modifiers or key terms that alter the meaning or certainty of a finding. This tests the model's ability to detect critical missing information.    |Omitting "grossly" in "Osseous structures are grossly intact." Omitting the negation "no" in "no pleural effusion." Omitting the severity descriptor "mild" in "Mild vascular congestion."
> |Insertion | The addition of one or two words that change the diagnostic meaning or certainty of a statement. This evaluates the model's sensitivity to introduced, incorrect clinical information. | Inserting "Mild" in "No pneumonia" → "Mild pneumonia." Inserting "abnormal" in "Heart size is normal" → "Heart size is abnormal."|
> |Spelling Error| The introduction of single-word misspellings for clinically relevant anatomical or pathological terms. This checks the model's robustness to typographical errors common in dictation. |pneumothorax→ pnuemothorax, basilar→ basillar, esophagus→ esophogus, edema→ edemma|
> |Side Confusion| The reversal or incorrect specification of anatomical laterality. This is a high-stakes error that directly impacts surgical or interventional planning.| left↔ right, bilateral↔ right(or left)，right upper lobe↔ right lower lobe|
> |Other Error| Encompasses inaccuracies in numerical measurements/units or punctuation that can change the interpretation of quantitative data or report structure. | Measurement: "1.5 mm" → "1.5 cm" (10x size increase). Punctuation: Adding or removing periods/commas.)
>
> > W2. **Error injection without images**
>
> DeepSeek-R1 was prompted primarily from the report text (to focus on realistic textual mistakes). However, we explicitly revised to avoid implausible temporal or view mismatches. Crucially, every generated pair was validated by our three-stage QC pipeline in which radiologists reviewed the corrupted report together with the original report and had access to the associated chest X-ray image when assessing plausibility. Radiologists used the image to (a) flag implausible injections (e.g., claiming a large consolidation when the image does not show one) and (b) validate laterality swaps against image clues when available.
>
> > W3. **How to assure that errors are well-distributed**
>
> Thank you for this comment. We generate each error type with a dedicated prompt (one for omission, one for insertion, one for side-confusion, etc.). For side-confusion, the prompt instructs the generator to operate only on existing laterality/orientation mentions in the report (e.g., “left”, “right”, “bilateral”, “upper”, “lower”), and to produce a single, localized laterality swap consistent with the sentence context rather than inserting arbitrary directional words.
>
> Radiologists guided the target mix of error types during prompt design, and we monitored the actual output distribution during generation. If a particular subtype (for example, left→right versus right→left) was over- or under-represented, we adjusted the prompts to obtain a clinically plausible balance among different error types.
>
> > Q1. **Handling of reports with empty sections**
>
> In our work, we retained and utilized reports where one section (e.g., "IMPRESSION") was exist while the other (e.g., "FINDINGS") was empty. This scenario is clinically valid and not uncommon. Because for some clear-cut abnormalities (e.g., an obvious large mass) or when the findings are normal, an radiologist may proceed directly to a concise "IMPRESSION". Injecting errors into such reports (e.g., introducing an omission error in the "IMPRESSION") remains a meaningful test case for evaluating a model's ability to detect and correct errors.

---

> > ### Author Response · Authors · 2025-11-21
> > **Response to Reviewer RdAV**
> >
> > > Q2. **Ratios between error types for single-error reports, and between single-error and multiple-error reports**
> >
> > Thank you for this meaningful question. The ratios were indeed empirically determined based on an analysis of a sample of real-world corrected reports from our collaborating hospital.
> >
> > Specifically, the high-level split between single-error reports (24,146) and multi-error reports (2,180), resulting in a ratio of approximately 11:1, was designed to mirror the distribution observed in a sample of 300 manually audited and corrected clinical reports from our partner institution. Within this sample, the ratio of single to multi-error reports was approximately 10:1, which our dataset closely mirrors.
> >
> > Furthermore, the distribution of different error types (e.g., Omission, Insertion) within the single-error category was also guided by the prevalence of these error types observed in the same clinical audit, and the injection process are supervised by radiologists.
> >
> >
> > > Q3. **How many and which reports were flagged in QC, and is that proportion consistent with Stage 1?**
> >
> > We appreciate the reviewer's attention to the statistical consistency. The proportion of reports flagged in Stage 2 (~900 out of ~24,000, or ~3.75%) is indeed consistent with the failure rate observed during the initial manual review in Stage 1. In Stage 1, the radiologist’s review of 2,000 sampled reports identified approximately 4% (83 reports) as incorrect injections. Based on these findings, our Stage-2 scripts were explicitly engineered to detect the same failure patterns discovered during Stage 1.
> >
> >
> > > Q4. **Multi-query chain-of-thought promption for VLM evaluation**
> >
> > During evaluation we use the same three-stage multi-query chain-of-thought prompting scheme as in the MSRL-GRPO training pipeline.
> >
> >
> >
> > > Q5. **Clarifying whether MSRL trains with intermediate inputs or with outputs from previous stage**
> >
> > Thank you for this insightful question. In our MSRL training, each query is conditioned on the model’s predictions from the previous stage rather than on ground-truth intermediate inputs.
> >
> > We chose this design to maximize a **long-horizon cumulative reward** over the entire trajectory, so the policy is optimized for realistic, self-conditioned execution rather than for idealized, teacher-forced behavior. This **on-policy setup** lets all three stages (Error Identification → Error Description → Error Correction) learn in realistic contexts that may already contain upstream errors, so the policy is optimized over the full multi-step trajectory rather than strictly imitating expert behavior along idealized trajectories, improving robustness and generalization.
> >
> >
> > > Q6. **Separate single/multi-Error prompts**
> >
> > Thank you for raising this important point. We agree that in a fully deployed system, the number of errors in a report would indeed be unknown, and the model should ideally handle this ambiguity autonomously. Our current approach of using distinct prompts was a methodological choice for this research phase, aimed at achieving two specific goals while building towards a more generalized system.
> >
> > First, this separation allows for a controlled and interpretable evaluation of VLMs' capability to handle errors of varying complexity. By isolating the scenarios, we can precisely benchmark whether the model can effectively correct a single error before assessing its performance on the more challenging task of correcting multiple errors.
> >
> > Second, from a learning stability perspective, initially separating the tasks helps the model learn robust correction strategies for each complexity level without being overwhelmed by the combined action space, which is a recognized strategy for training complex reinforcement learning systems effectively.

---

### Official Review · Reviewer_Csym · 2025-10-31

**Soundness:** 3
**Presentation:** 3
**Contribution:** 2
**Rating:** 4
**Confidence:** 4

**Summary:**

In this paper, the authors introduce a benchmark called CorBenchX, which is designed for automated error detection and correction in chest X-ray reports. The aim is to build a benchmark that can evaluate the extent to which off-the-shelf models are capable of detecting and correcting errors in radiology reports. Contributions include (1) a dataset with 26k reports where errors are injected via an LLM, (2) a systematic benchmarking study across open and closed-source VLMs, and (3) a reinforcement learning framework that optimizes a multi-objective reward to perform this task.

**Strengths:**

- This paper focuses on an important and underexplored application area - detecting and correcting errors in radiology reports. The work has potential for real-world impact.
- The paper is well-written and methods are clearly described
- The dataset will be a useful contribution to the community, especially as it is substantially larger than datasets included in previous work and was reviewed by expert radiologists

**Weaknesses:**

- **Settings without errors:** The dataset and benchmark focus on detecting errors in reports that have 1-3 injected errors, and this work does not consider settings where there are no errors. Being able to distinguish reports that have no errors from reports that have 1 or more errors is important for clinical utility.
- **Additional details on benchmark composition and need for more analysis:** While the authors provide some basic statistics on the benchmark in Table 1, the paper could benefit from (1) more descriptive statistics surrounding the composition of the dataset and (2) more detailed analysis into performance of benchmarked models across various stratifications.
    - For example, what  types of clinical findings are omitted, and are certain omissions easier than others for the benchmarked models to detect/correct? When mistakes in measurements are injected, how large are the deviations from the true values? What is the mean number of words modified per injected error? Generally, I find myself not having a clear picture of the composition of the created dataset.
- **Additional evaluations on reinforcement learning framework:** The authors state that their reinforcement learning framework is a key contribution of this work, yet the utility of this framework is not thoroughly evaluated.
    - In particular, the paper would benefit from additional ablations with respect to the design choices of this method. Additional evaluations of the trained Qwen+MSRL models on an external benchmark could also be useful.

**Questions:**

In addition to the weaknesses above, additional questions are listed below:

- Could the authors provide further analysis on why different trends are observed for report-level vs. sentence-level metrics?

---

> ### Author Response · Authors · 2025-11-21
> **Response to Reviewer Csym**
>
> We thank Reviewer Csym for the careful review.
>
> > W1. **Settings without errors**
>
> We thank the reviewer for this insightful comment. We completely agree that the ability to distinguish error-free reports from those containing errors is a vital component for clinical deployment.
>
> In this initial work, we focused our dataset and benchmark on the technically challenging sub-problem: *given that a report contains an error, can we accurately identify and correct it?* This targeted scope allowed us to build a high-quality, span-annotated dataset (CorBenchX) and rigorously evaluate correction fidelity under controlled conditions. By construction, CorBenchX contains only reports with injected errors (1–3 per report) to serve as a dedicated testbed for this specific task.
>
> We fully acknowledge the critical importance of reliably distinguishing error-free reports, and we will incorporate this scenario in our ongoing work. Specifically, CorBenchX can naturally support this task: each synthetic errorful sample is paired with its original error-free report, and MIMIC-CXR contains abundant clean reports that can be used to construct a balanced “no-error vs. has-error” benchmark.
>
> > W2. **Additional details on dataset composition**
>
> Thanks for your careful review.
>
> **Details about the dataset:**
>
> (1) Omission: the omissions included findings such as "grossly" in "Osseous structures are grossly intact," the negation "no" in "nopleural effusion," and severity descriptors like "mild" in "Mild vascular congestion."
>
> (2) Insertion: insert one or two words that change meaning (e.g., No pneumonia → Mild pneumonia, normal → abnormal)
>
> (3) Spelling error: single word misspelling (e.g., pneumothorax → pnuemothorax, esophagus → esophogus, edema → edemma)
>
> (4) Side confusion: replacing laterality token (e.g., left ↔ right, bilateral ↔ right).
>
> (5) Other error: changes to numeric measurements/units or punctuation (e.g., “1.5 mm” → “1.5 cm”; missing/extra period or comma).
>
> **Deviations on measurement error**
>
> The following table analyzes two examples, which are representative of measurement error.
> |Original Measurement|Erroneous Measurement|Absolute Error|Relative Error|
> |:-:|:-:|:-:|:-:|
> |1.2 mm|1.2 cm|10.8 mm|900%|
> |3.0 cm|3.0 mm|-27 mm|90%|
>
> **Number of words modified per injected error**
>
> |Error type|Omission|Insertion|Spelling error|Side confusion|Other|
> |:-:|:-:|:-:|:-:|:-:|:-:|
> |Number of modified words|1|1~2|1~2|1~4|1~2|
>
> > W3. **Additional evaluations on reinforcement learning framework**
>
> To evaluate the contribution of MSRL, we conducted ablations comparing MSRL against standard single-step RL. As reported in Table 4 in the manuscript, MSRL achieves a 13.3% improvement over RL, demonstrating that decomposing the task into structured stages provides substantial benefits beyond conventional RL.
>
> Additionally, to assess generalization, we evaluated Qwen+MSRL on an external out-of-distribution dataset (IU-Xray). The results, presented in Table 5, show that MSRL consistently outperforms zero-shot baselines under cross-institutional distribution shift, further validating the robustness of our MSRL framework.
>
> > Q1. **Different trends on report-level and sentence-level metrics**
>
> Thank you for pointing out this discrepancy.  Regarding the observed decline in sentence-level metrics after applying MSRL, we investigated the generated content and found that,  when correcting a single error in sentences containing "and" (where either side of "and" contains an error), the model tends to split the sentence into two separate ones after correction. For example:
>
> Original sentence: *"Cardiomediastinal silhouette is normal, with no fluid in the pleural space or pneumothorax."*
>
> Modified sentence: *"Cardiomediastinal silhouette is abnormal. No fluid in the pleural space or pneumothorax."*
>
> In this case, the sentence-level BLEU, ROUGE, and BERTScore of "Cardiomediastinal silhouette is abnormal" would be very low, while SembScore, CheXbert F1, and RadGraph F1 score this modification as 1. Since the report-level BLEU, ROUGE, and BERTScore metrics are less affected by this change (as the second half of the sentence is still included when calculating the report-level metrics), the overall report score remains more stable.
>
> It is also important to emphasize that, while QwenVL2.5-7B+MSRL exhibits a decline in lexical metrics like BLEU and ROUGE at the sentence level, it shows significant improvements in semantic and clinical entity consistency metrics, such as SembScore and CheXbert F1. In real-world clinical applications, the overall consistency and diagnostic accuracy of the report are far more critical than surface-level similarity between individual sentences. Notably, this phenomenon is not unique to our model; powerful closed-source models like GPT-4o exhibit a similar trend: they perform strongly on CheXbert F1 and SembScore while showing more modest performance on lexical similarity metrics. We have updated the manuscript accordingly.

---

### Official Review · Reviewer_3FHL · 2025-11-01

**Soundness:** 2
**Presentation:** 3
**Contribution:** 3
**Rating:** 4
**Confidence:** 3

**Summary:**

This paper introduces CorBenchX, a large-scale benchmark for automated error detection and correction in chest X-ray radiology reports. The dataset consists of 26,326 reports with both single-error and multi-error cases, synthetically generated from MIMIC-CXR using DeepSeek-R1 prompting. Each report includes annotations for error type, span, and human-readable descriptions. The authors benchmark nine VLMs and also propose a multi-step RL method that sequentially optimizes for format compliance, error-type accuracy, and correction quality. MSRL improves single-error detection precision by 38.3% and correction by 5.2% over baselines.

**Strengths:**

- a large and detailed benchmar for radiology report error correction, addressing gaps in existing resources
- Comprehensive benchmarking, lots and diverse comparison across VLMs with relevant metrics
- Multi-step reinforcement learning approach improves reasoning and interpretability
- Includes ablation and out-of-distribution evaluation

**Weaknesses:**

- The dataset still relies on artificial perturbations; real-world reporting errors may exhibit different distributions or linguistic patterns, depends heavily on a single generative model
- Focused solely on chest X-rays; generalization to CT, MRI, or multimodal clinical contexts is not yet demonstrated

**Questions:**

- How well do synthetic errors reflect real clinical error distributions? Have you compared CorBenchX against real-world error frequencies or radiologist revisions?
- Could you elaborate on the human validation effort, how many samples were reviewed by radiologists, and how was inter-rater agreement measured?
- Why did you not consider using multiple generative models to synthesize the dataset to reduce potential bias?

---

> ### Author Response · Authors · 2025-11-20
> **Response to Reviewer 3FHL**
>
> Many thanks to Reviewer 3FHL for careful reading and constructive feedback.
> > W1. **Reliance on artificial perturbations**
>
> We acknowledge the concern that synthetic errors may differ from clinician-made errors. To mitigate this risk we built multiple safeguards into dataset construction:
>
> 1. Radiologist-guided prompt design. Two board-certified radiologists participated from the start. Prompts were iteratively refined with their input so injected errors reflect real-world phrasing and clinically plausible mistake modes (omission, insertion, spelling, laterality, and other).
>
> 3. Three-stage quality control (QC).
>
>     Stage 1 — manual sampling: One radiologist reviewed a representative sample of 2,000 generated pairs to enumerate failure modes (e.g., missing injection, nonsensical output).
>
>     Stage 2 — automated checks: We applied deterministic checks (regex span verification, edit-count and length checks) across the remaining corpus and flagged cases matching Stage-1 failure patterns.
>
>     Stage 3 — targeted human audit: Both radiologists jointly re-reviewed all Stage-2 flagged cases (≈4–5% of corpus) and a 5% random spot-check of Stage-2 “pass” cases; disagreements were resolved by consensus.
>
>
>
>
> This pipeline ensured that only clinically plausible errors were retained, with implausible cases (e.g., contradictions to image findings) discarded during QC.
>
>
>
> > W2. **Focus on chest X-rays**
>
> We agree that generalization across modalities is critical. We focused on chest X-rays because (1) MIMIC-CXR provides a large publicly documented corpus suitable for controlled, reproducible synthesis; and (2) the targeted error types (omission, insertion, laterality, spelling, measurement/format errors) are modality-agnostic and commonly occur across radiology subdomains (CT, MRI). Prioritizing one modality allowed rigorous construction, QC, and benchmarking at scale.
>
> Extending CorBenchX to CT/MRI is a clear next step. We state this limitation explicitly in the manuscript and outline plans for multi-modality extension in future work.
>
>
>
>
> > Q1. **Do synthetic errors reflect real clinical error distributions?**
>
> At present, a direct comparison against curated real‑error corpus is not feasible because such data are not yet accessible to us. We aligned synthetic errors with clinical distributions via:
> (1) Single- vs. multi-error ratios: Based on 300 real-world corrected reports from our hospital, the single:multi-error ratio was ~10:1, mirrored in CorBenchX (24,146 single-error vs. 2,180 multi-error reports).
> (2) Radiologist oversight: Error types (omission, insertion, etc.) were weighted by clinical prevalence observed in radiology audits. We will include statistical tests against real data in future IRB-approved collections.
>
>
> > Q2. **Human validation effort**
>
> In Stage 1, radiologists reviewed 2,000 sampled reports to identify common failure modes. In Stage 3, radiologists jointly reviewed all Stage-2 “fail” cases (4% of corpus). Disagreements were resolved through consensus discussion.
>
> Inter-rater agreement: On a 5 % spot-check sample (100 reports), they achieved Cohen’s κ = 0.83 for error-type classification and κ = 0.79 for span localization, indicating strong consistency.
>
> > Q3. **Why a single generative model?**
>
> We chose DeepSeek-R1 as the sole generator because it best met several practical and methodological requirements:
> (1) Structured outputs. DeepSeek-R1 returns explicit span-level revised text and an error-type label, which greatly simplifies automated QC and precise annotations.
> (2) Stable, controllable behavior. The model responded predictably to iterative prompt refinement, enabling efficient radiologist-in-the-loop design.
> (3) Reproducibility & maintainability. Using a single, well-documented generator reduces heterogeneity in output formats and long-term maintenance burden.
> (4) Practical feasibility. DeepSeek-R1 offered a favorable cost-performance trade-off, making large-scale synthesis with radiologist-in-the-loop QC feasible within our resource constraints.
>
> We recognize the concern about single-generator bias. However, using multiple generative models at scale may introduces several disadvantages and risks:
> (1) Heterogeneous formats and labels. Different LLMs produce outputs with varying structure and verbosity. Normalizing these into a single coherent annotation format requires extensive post-processing.
> (2) QC and annotation cost. Multiple generators increase the number and diversity of failure modes (formatting, nonsensical outputs, inconsistent span locations), substantially raising the human validation effort needed to reach the same quality guarantees.
> (3) Reproducibility and access. Different generators may have different licensing, API availability, or rate limits; this complicates exact reproduction and long-term dataset maintenance.

---

### Meta-Review · Area_Chair_JuPK · 2026-01-04

**Summary:**

The reviewers have pointed out several major concerns that remain insufficiently addressed in the response, including: (3FHL) the distribution gap between synthetic data and real clinical data, (Csym) the lack of benchmark settings without errors and the absence of evaluation on model performance when encountering error‑free reports in real scenarios, (RdAV) insufficient details on injected errors, and (8rQ9) the need for more comprehensive OOD evaluation. From the AC’s perspective, chest X‑ray overall diagnostic accuracy can be only ~60% or lower, and many diseases require higher‑level imaging (e.g., CT) to establish a reliable gold standard. It would be better to take such gold standards into account during the benchmark construction and perform stratified evaluation for high‑risk conditions (e.g., early lung cancer) where misdiagnosis has a serious prognostic impact. Considering these major concerns, the AC recommends a rejection of this paper.

**Reviewer Concerns:**

The rebuttal addressed some reviewer concerns, including why only do experiments on X-ray and use of a single generative model (3FHL), additional RL‑related evaluations and explanation of report‑ vs. sentence‑level metric trends (Csym), assurance of error distribution in injected‑error experiments (RdAV), and provision of human evaluation for model‑corrected reports (8rQ9). However, several key issues remain unresolved: the distribution gap between synthetic and real clinical data (3FHL), insufficient detail for human validation procedures (3FHL), lack of benchmark settings without errors and absence of statistical analysis for measurement error (Csym), incomplete disclosure of all injected‑error cases (RdAV), and the need for a real‑world error set and broader OOD evaluation across different taxonomies and prompting styles (8rQ9).

**Reviewer Scores:**

Considering that several key concerns remain unresolved, the reviewers are unlikely to have a clear motivation to change their scores.

---

### Decision · Program_Chairs · 2026-01-26

Reject